

# Combining legacy data with new drone and DGPS mapping to identify the provenance of Plio-Pleistocene fossils from Bolt's Farm, Cradle of Humankind (South Africa)

Tara R. Edwards[1], Brian J. Armstrong[1], Jessie Birkett-Rees[2], Alexander F. Blackwood[1], Andy I.R. Herries[1,3], Paul Penzo-Kajewski[1], Robyn Pickering[4,5] and Justin W. Adams[3,6]

[1] The Australian Archaeomagnetism Laboratory, Department of Archaeology and History, La Trobe University, Melbourne, Victoria, Australia
[2] Centre for Ancient Cultures, Faculty of Arts, Monash University, Clayton, Melbourne, Victoria, Australia
[3] Centre for Anthropological Research, University of Johannesburg, Johannesburg, Gauteng, South Africa
[4] Department of Geological Science, University of Cape Town, Cape Town, Western Cape, South Africa
[5] Human Evolution Research Institute, University of Cape Town, Cape Town, Western Cape, South Africa
[6] Centre for Human Anatomy Education, Department of Anatomy & Developmental Biology, Biomedical Discovery Institute, Faculty of Medicine, Nursing & Health Sciences, Monash University, Clayton, Melbourne, Victoria, Australia

Corresponding author
Tara R. Edwards,
T.Edwards@latrobe.edu.au

## ABSTRACT

Bolt's Farm is a Plio-Pleistocene fossil site located within the southwestern corner of the UNESCO Hominid Fossil Sites of South Africa World Heritage Site. The site is a complex of active caves and more than 20 palaeokarst deposits or pits, many of which were exposed through the action of lime mining in the early 20th century. The pits represent heavily eroded cave systems, and as such associating the palaeocave sediments within and between the pits is difficult, especially as little geochronological data exists. These pits and the associated lime miner's rubble were first explored by palaeoanthropologists in the late 1930s, but as yet no hominin material has been recovered. The first systematic mapping was undertaken by Frank Peabody as part of the University of California Africa Expedition (UCAE) in 1947–1948. A redrawn version of the map was not published until 1991 by Basil Cooke and this has subsequently been used and modified by recent researchers. Renewed work in the 2000s used Cooke's map to try and relocate the original fossil deposits. However, Peabody's map does not include all the pits and caves, and thus in some cases this was successful, while in others previously sampled pits were inadvertently given new names. This was compounded by the fact that new fossil bearing deposits were discovered in this new phase, causing confusion in associating the 1940s fossils with the deposits from which they originated; as well as associating them with the recently excavated material. To address this, we have used a Geographic Information System (GIS) to compare Peabody's original map with subsequently published maps. This highlighted transcription errors between maps, most notably the location of Pit 23, an important palaeontological deposit given the recovery of well-preserved primate crania (*Parapapio*, *Cercopithecoides*) and partial skeletons of the extinct felid *Dinofelis*. We conducted the first drone and Differential

Global Positioning System (DGPS) survey of Bolt's Farm. Using legacy data, high-resolution aerial imagery, accurate DGPS survey and GIS, we relocate the original fossil deposits and propose a definitive and transparent naming strategy for Bolt's Farm, based on the original UCAE Pit numbers. We provide datum points and a new comprehensive, georectified map to facilitate spatially accurate fossil collection for all future work. Additionally, we have collated recently published faunal data with historic fossil data to evaluate the biochronological potential of the various deposits. This suggests that the palaeocave deposits in different pits formed at different times with the occurrence of *Equus* in some pits implying ages of <2.3 Ma, whereas more primitive suids (*Metridiochoerus*) hint at a terminal Pliocene age for other deposits. This study highlights that Bolt's Farm contains rare South African terminal Pliocene fossil deposits and creates a framework for future studies of the deposits and previously excavated material.

## INTRODUCTION

Bolt's Farm is the name given to a series of fossil bearing palaeocave remnants located ~1.5–3.0 km to the southwest of the early Pleistocene early hominin (*Paranthropus robustus,* early *Homo* and *Australopithecus africanus*) bearing sites of Swartkrans and Sterkfontein, and ~1 km south of the Rising Star Cave system (*Homo naledi*) (*Berger et al., 2015*; *Dirks et al., 2015*) (Fig. 1). Apart from the little explored archaeological and fossil bearing site of Goldsmith's (*Mokokwe, 2007*) 0.5 km to the south, Bolt's Farm is the most southwestern fossil-bearing site in the Gauteng exposures of the Malmani dolomite UNESCO Hominid Sites of South Africa World Heritage Site (colloquially referred to as 'The Cradle'). The pits and caves that are now collectively referred to as Bolt's Farm occur on three properties: the western Klinkerts property, the eastern Greensleeves Property, and the northern Sterkfontein Quarry (Fig. 2). The fossil site is named after Mr Billy Bolt, the owner of the original farm that sat on the eastern Greensleeves property and Sterkfontein Quarry (known as Main Quarry). The western Klinkerts part of the site was owned by the Clyde Trading Company (indicated on the original site map as the Amlors Ors Co.; SOM SF1, SF2).

As with the other caves in the area, Bolt's Farm was heavily mined for speleothem (calcium carbonate from stalagmites, stalactites and flowstones) in the terminal 19th and early 20th centuries. The speleothem was burnt in kilns to make lime for use in the gold extraction process. Evidence for this is preserved as lime miner's cottages and kilns that survive at both the northeast and southeastern end of the Greensleeves Property (Fig. 2). While discrete deposits existed, mining revealed and created a series of pits and dumps from which fossils were collected from the 1936 (*Broom, 1937*), to the current projects (*Pickford & Gommery, 2016*).

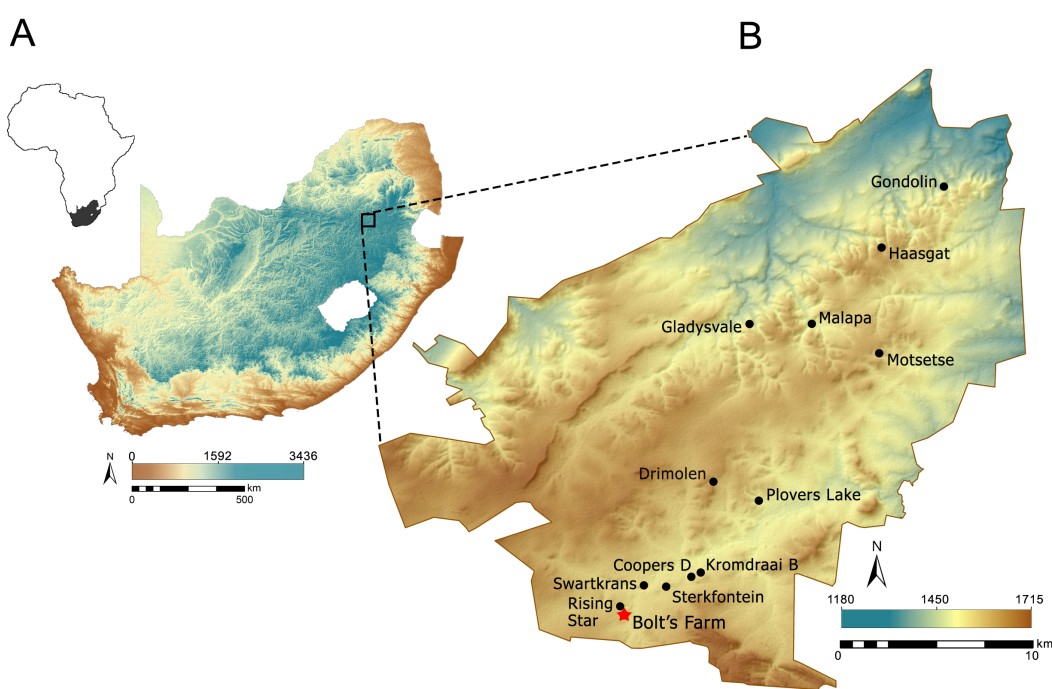

**Figure 1** Location of the Cradle in South Africa (A) and Bolt's Farm within the Cradle (B). Elevation data made available from *Jarvis et al. (2008)*.

The significance of Bolt's Farm lies both within this numerous, extensive network of pits that have yielded a diverse range of faunal material (SOM Text S1) and the suggested Pliocene ages for some of the specimens (*Sénégas & Avery, 1998*; *Gommery et al., 2008a*). Early mentions described Bolt's Farm as a single deposit (*Cooke, 1963*), while later work recognised the inherent complexity and published faunal data relating to specific pits (e.g., *Delson, 1984*; *Cooke, 1991*; *Cooke, 1993*). It is now generally accepted that the site consists of deposits of various ages that formed either as part of the same cave system at different times (*Gommery et al., 2012*), or may represent the infill of several completely unconnected caves. Although several publications have used biochronological correlations to suggest depositional ages for specific pits at Bolt's Farm (e.g., *Delson, 1984*; *Sénégas & Avery, 1998*; *Reynolds, 2007*; *Gommery et al., 2008a*), no comprehensive review of the biochronologically sensitive taxa has been attempted. Recent Cradle-wide dating suggests some cave localities may be younger than previously thought (*Pickering et al., 2018*), which has particular impact on biochronological interpretations of some Bolt's Farm pits forming within the earlier Pliocene (*Sénégas & Avery, 1998*; *Gommery, Sénégas & Thackeray, 2008b*).

While the use of spatial aids e.g., geographic information system (GIS), remote sensing and photogrammetry for visualising landscapes has a strong history in archaeology (*Gibbons, 1991*; *Lock & Stancic, 1995*; *Birkenfeld, Avery & Horwitz, 2015*; *De l Del la Torre et al., 2015*; *Fernández-Lozano & Gutiérrez-Alonso, 2016*; *Jorayev et al., 2016*; *Dell'Unto et al., 2017*) its application to palaeoanthropology and palaeontology has previously been acknowledged as lagging (*Conroy et al., 2008*; *Anemone, Conroy & Emerson, 2011*). These

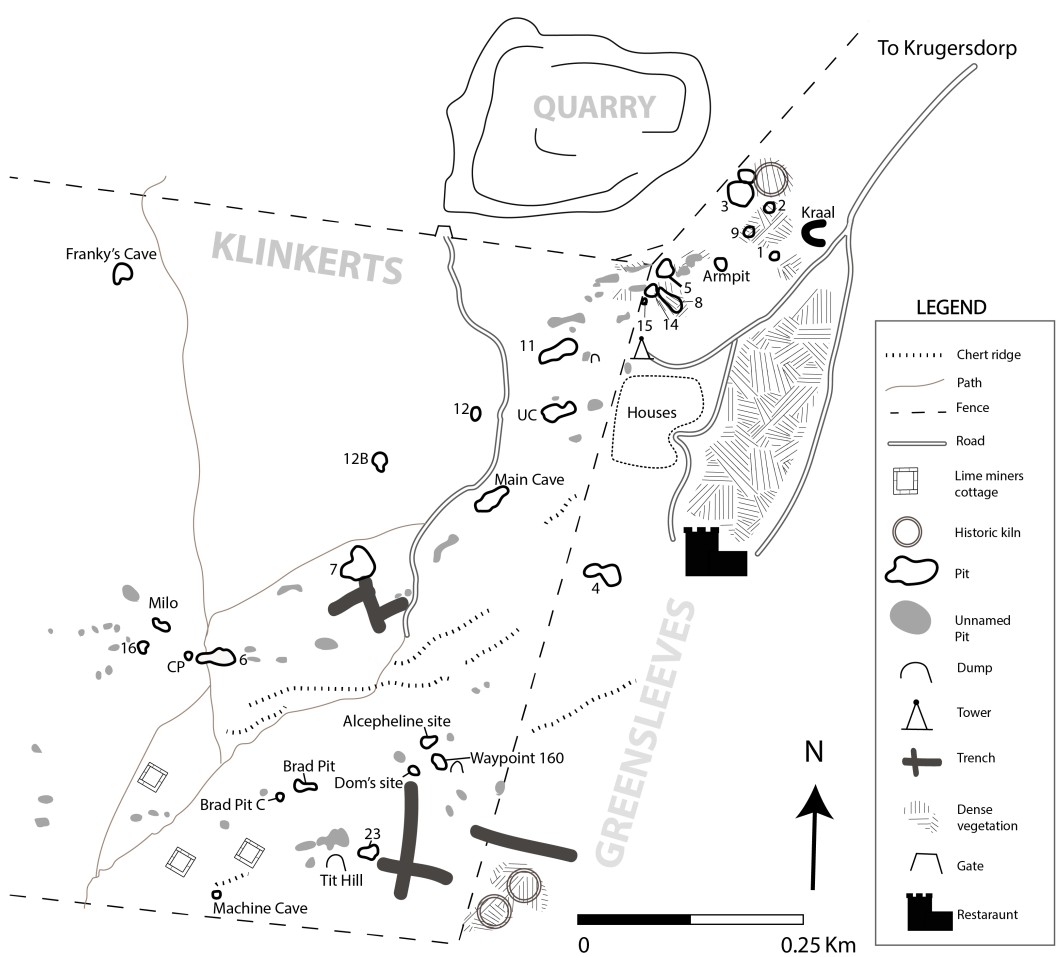

**Figure 2 New georectified map of Bolt's Farm from accurate DGPS survey.** Coordinate system WGS1984 UTM35S.

methods are wide reaching and can be applied on a landscape scale e.g., mapping and identifying fossil bearing outcrops (*Oheim, 2007*), mapping hominin migration routes (*Holmes, 2007*) and reconstructing palaeoenvironments (*Bailey, Reynolds & King, 2011*). GIS is also extremely valuable on a intra-site scale, allowing not only for visualisation (*Armstrong et al., 2018*) but analysis and reconstruction of bone and stone accumulations (*Nigro et al., 2003*). While highly valuable, these methods have yet to be applied to Bolt's Farm.

In this contribution, we chronicle the previous work carried out on the Bolt's Farm pits, from the 1930s to the present, with a particular focus on the names and locations of the various deposits (Table 1). To this end, we provide new spatial data and make available accurate survey control points for future use (SOM SF3). The aim of this is to reduce the confusion regarding pit location and naming, which are the result not only of staggered research since the early 20th century but the intrinsically complex nature of the deposits across the surface at Bolt's Farm. We also present an overview of the previously described

**Table 1  Known locations across Bolt's Farm and various names within the literature, sorted by source.** Coordinates from DGPS survey given in South African Grid and UTM.

| F Peabody (1947, unpublished data) | UCMP Locality | Cooke (1991) | Sénégas et al. (2002) | Thackeray et al. (2008) | Zipfel & Berger (2009) | Gommery et al. (2012) | Monson, Brasil & Hlusko (2015) | Pickford & Gommery (2016) | This publication | SA Hartebeesthoek 94/ Lo27 | UTM -35 |
|---|---|---|---|---|---|---|---|---|---|---|---|
| Pit 1 (Kraal Pit) | V67256, V75133 | Pit 1 Kraal Pit | Pit 1 | Kraal Cave | Kraal Pit (Pit 1) | Pit 1 | Pit 1 (Kraal Pit) | Pit 1 | Pit 1 | −71816.550Y 2880218.092X | 7120933.995N 571787.823E |
| Pit 2 (Kiln Cave) | V67257 | Pit 2 Kiln | Pit 2 | H Cave | H Cave (Pit 2) | N/A | Pit 2 (Kiln Pit) | H Cave | Pit 2 | −71808.454Y 2880137.641X | 7121014.414N 571779.731E |
| Pit 3 (KB Cave) | V67258, V75132 | Pit 3 KB Cave | Pit 3 | Cobra Cave | KB/Cobra Cave (Pit 3) | Cobra Cave | Pit 3 (Cobra Cave) | Cobra Cave | Pit 3 (Cobra Cave) | −71775.725Y 2880150.923X | 7121001.137N 571747.015E |
| Pit 4 (Garage Ravine) | V67259 | Pit 4 Garage Ravine Cave | Pit 4 | Garage Ravine Cave | Garage Ravine Cave (Pit 4) | Garage Ravine Cave | Pit 4 (Garage Ravine Cave) | Garage Ravine | Pit 4 | −71623.214Y 2880568.009X | 7120584.218N 571594.565E |
| Pit 5 (Smith Cave) | V67260, V75139 | Pit 5 Smith Cave | Pit 5 | Smith Cave-misidentified | Smith Cave (Pit 5) | Smith Cave | Pit 5 (Smithy Cave) | Aves Cave 4 (listed as Pit 13) | Pit 5 | −71692.381Y 2880228.869X | 7120923.223N 571663.704E |
| Pit 6 (Baboon Cave) | V67261 | Pit 6 Baboon Cave | Pit 6 | Baboon Cave | Baboon Cave (Pit 6) | Baboon Cave | Pit 6 (Baboon Cave) | Baboon Cave | Pit 6 | −71196.127Y 2880661.711X | 7120490.554N 571167.649E |
| Pit 7 (Elephant Cave) | V67262 | Pit 7 Elephant Cave | Pit 7 | Bridge Cave | Elephant/ Bridge Cave (Pit 7) | Bridge Cave | Pit 7 (Elephant Cave) | Bridge Cave | Pit 7 | −71348.713Y 2880563.021X | 7120589.204N 571320.174E |
| Pit 8 | V75269 | Pit 8 | N/A | Rodent Cave | Rodent Cave (Pit 8) | Rodent Cave | Pit 8 (Rodent Cave) | Aves Cave 2 | Pit 8 | −71700.181Y 2880266.450X | 7120885.656N 571671.501E |
| Pit 9 | N/A | Pit 9 | Pit 9 | No name | No name (Pit 9) | N/A | N/A | Pit 9 | N/A | −71790.951Y 2880193.79X | 7120958.288N 571762.235E |
| Bushman Outcrop | N/A | Breccia outcrop | Breccia Outcrop | Breccia Outcrop | N/A | Milo A | N/A | Milo A | Milo | −71131.98Y 2880625.805X | 7120526.445N 571103.527E |
| Pit 10 | V67263 | Pit 10 Grey Bird Pit | N/A | Main Quarry | Grey Bird Pit/Main Quarry (Pit 10) | N/A | N/A | N/A | N/A | Destroyed – approx loc. -71810.363Y 2880123.234X | 7121028.815N 571781.639E |
| Pit 11 | N/A | Pit 11 | Pit 11 | U Cave | N/A (Pit 11) | X Cave | N/A | X Cave | Pit 11 | −71569.186Y 2880320.273X | 7120831.855N 571540.558E |
| Pit 12 | N/A | Pit 12 | Pit 12A | No name | No name (Pit 12A) | Pit 12 (A) | N/A | Pit 12 (A) | Pit 12 | −71487.209Y 2880393.871X | 7120758.287N 571458.614E |
| N/A | N/A | N/A | Pit 12B | N/A | No Name (Pit 12B) | Pit 12B | N/A | Pit 12 b | Pit 12B | −71377.978Y 2880444.538X | 7120707.640N 571349.426E |
| Pit 13 | N/A | Pit 13 | Pit 13-Misidentified (Pit 5 was mapped) | Arm Pit | (Pit 13) | N/A | N/A | Aves Cave 5 | N/A | −71684.94606Y 2880222.8518X | 7120929.237N 571656.272E |

**Table 1** (*continued*)

| F Peabody (1947, unpublished data) | UCMP Locality | *Cooke (1991)* | *Sénégas et al. (2002)* | *Thackeray et al. (2008)* | *Zipfel & Berger (2009)* | *Gommery et al. (2012)* | *Monson, Brasil & Hlusko (2015)* | *Pickford & Gommery (2016)* | This publication | SA Harte- beesthoek 94/ Lo27 | UTM -35 |
|---|---|---|---|---|---|---|---|---|---|---|---|
| Pit 14 (Bench mark Pit) | V67264 | Pit 14 Bench- mark Pit | Pit 14 | Benchmark Pit | Bench Mark Pit (Pit 14) | Benchmark Pit | Pit 14, Bench- mark Pit, Lo- cation 10 | Aves Cave 1 | Pit 14 | −71680.196Y 2880248.291X | 7120903.808N 571651.524E |
| Pit 15 | V73105 | Pit 15 | Pit 15- Misidentified | Aves Cave | Aves Cave (Pit 15) | Aves | Pit 15, Aves, Location 11 | Aves Cave 6 | Pit 15 | −71671.637Y 2880262.266X | 7120889.838N 571642.968E |
| Pit 16 (Equine Pit) | V67265 | Pit 16 Equine Pit- cut off map | N/A | N/A | N/A | Milo B | N/A | Milo B | Pit 16 | −71109.010Y 2880649.901X | 7120502.359N 571080.566E |
| Pits 17–22 | N/A | Not mapped | N/A | N/A | N/A | N/A | N/A | N/A | N/A | N/A | N/A |
| Pit 23 | V4888 | Pit 23 Tit Hill Pit | Pit 23- Misidentified | Tit Hill Pit - Misidentified | Tit Hill Pit (Pit 23) | Tit Hill Pit - Misidentified | Pit 23, Tit Hill Pit, Loca- tion 13 | Tit Hill Pit - Misidentified | Pit 23 (Tit Hill Pit) | −71363.419Y 2880879.361X | 7120272.991N 571334.874E |
| Tit Hill | V67270 | Old Dumps (*Cooke, 1991*) | Femur Dump | | N/A | Femur Dump | Pit 23, Bolts Farm Dump, Location 13 | Femur Dump | Tit Hill | −71326.245Y 2880884.057X | 7120268.297N 571297.715E |
| Pit 24 | N/A | N/A | N/A | N/A | N/A | N/A | N/A | N/A | N/A | No location data made available | No location data made available |
| Pit 25 (Gazelle Pit) | V67267 | Pit 25 (Gazelle Pit) | N/A | N/A | N/A | N/A | N/A | N/A | N/A | No location data made available | No location data made available |
| N/A | V67268 | N/A | N/A | N/A | N/A | N/A | New Cave | N/A | N/A | No location data made available | No location data made available |
| N/A | V67269 | N/A | N/A | N/A | N/A | N/A | Jackal Cave | N/A | N/A | No location data made available | No location data made available |
and undescribed faunal material reposited across US and South African institutions with the aim of providing key biochronological ages for the Bolt's Farm deposits where possible. In doing so we also provide the first basis for associating historic and more recently developed fossil samples excavated from these pits, a critical step in reconciling the faunal record from across this prolific locality and allowing for more justified intra- and intersite faunal, taphonomic and palaeoecological analyses.

## REVIEW OF PREVIOUS EXCAVATIONS, MAPPING AND NOMENCLATURE AT BOLT'S FARM

The first mentions of Bolt's Farm are by *Broom (1937)* but there is confusion as to the definite locality to which he is referring. *Broom (1937)* and *Broom (1939)* used a number of site location names no longer used today: referring interchangeably to 'Sterkfontein Farm', 'Sterkfontein Caves', 'Bolt's Farm' and 'Bolt's Workings at Sterkfontein'. In his initial publications, *Broom (1937)* and *Broom (1939)* described a number of novel carnivores *Leptailurus spelaeus* (Family Felidae, Order Carnivora; figured in *Broom (1939)* but specimen not currently locatable), *Crossarchus transvaalensis* (Family Herpestidae, Order Carnivora; figured in *Broom (1939)* but specimen not currently locatable), and the type specimen of the extinct hedgehog *Atelerix major* (Family Erinaceinae, Order Eulipotyphla; TM 1544; subsequently subsumed into *Erinaceus (Atelerix) broomi* per *Werdelin & Peigne, 2010*). These specimens are described as originating from "Sterkfontein in a cave, about a mile south of that in which *Australopithecus* was found" (*Broom, 1937* pp. 512), which fits the known location of what today is Bolt's Farm. *Broom (1939)* further qualifies the location of these specimens as "found at Bolt's workings on Sterkfontein" (*Broom, 1939* pp. 333) alongside the description of the STS 130-299 specimen *Machairodus transvaalensis* (Family Felidae: Order Carnivora). Broom continued to sample at Bolt's Farm until 1948, describing additional type specimens such as *Felis shawi* (BF 1555; Family Felidae, Order Carnivora; subsequently subsumed into *Panthera leo Linnaeus, 1758*) and *Elephantulus antiquus* (Family Macroscelididae, Order Macroscelidae; figured in *Broom (1948)* but specimen not currently locatable), as well as preserved remains of *Phacochoerus modestus* (BF3-3355; Family Suidae, Order Cetartiodactyla; subsequently subsumed into *Phacochoerus antiquus Broom, 1948*; *Adams et al., 2015*; see SOM Text S1). There has been considerable confusion over the provenance of these early fossil specimens to what is currently defined as Bolt's Farm, let alone specific pit deposits due to the ambiguity of these early reports that sadly likely cannot be addressed short of direct specimen sampling (e.g., *Trueman et al., 2005*).

Between 1947 and 1948, the southern section of the University of California Africa Expedition (UCAE) visited Bolt's Farm, led by C.L. Camp and F. E. Peabody (*Camp, 1948*). Their aim was to gain further fossil evidence and geological context for the australopithecine specimens described by *Dart (1925)* and *Broom (1936)*. The UCAE undertook systematic sampling of fossiliferous calcified deposits across the Cradle, including from several miners pits and rubble on Bolt's Farm. While members of the UCAE did keep detailed field dairies recording daily activities and discoveries, it is often difficult to reconcile whether

specimens were identified *in situ* or collected from miner's rubble. Further, some localities have several rubble dumps nearby and subsequently it can be difficult to associate a rubble dump with any one pit. Attention was often paid to the matrix adhering to any specimens collected, and attempts made to match this with sediment in a nearby locality. Frank Peabody created the first known map of the site (SOM SF1 SF2; list of pits Table 1), which was not published in its original form until recently (*Monson, Brasil & Hlusko, 2015*)—although used by *Cooke (1991)* to generate his map (see below). The expedition amassed a significant collection of fossils from a range of sites, now housed at the University of California Museum of Paleontology (UCMP) (*Peabody, 1954*; *Monson, Brasil & Hlusko, 2015*), with some specimens repatriated to Evolutionary Studies Institute at the University of the Witwatersrand (Johannesburg) and the Ditsong National Museum of Natural History (Pretoria), South Africa.

Due to his sudden death in 1958, Peabody was unable to prepare a detailed report of his work at Bolt's Farm, as he had done for Taung (*Peabody, 1954*). Subsequently, Cooke visited the UCMP in 1957–1958 (as well as in 1975 and 1983) to study the fossils recovered by the expedition (*Cooke, 1991*). *Cooke (1991* p.9) published a map "redrawn directly" from Peabody's survey map, including pit numbers, associated names and locality numbers from the UCAE (Pits 1–16 and 23–25).

The Palaeontological Expedition to South Africa (PESA) ran from 1996–1999 under the direction of Senut and Pickford (*Sénégas & Avery, 1998*). The project undertook further collections from fossil dumps and attempted to relocate all sites from the UCAE using *Cooke*'s (*1991*) map (*Sénégas et al., 2002*). While they were not able to identify all the sites with certainty, the project did discover a new site, Waypoint 160 (*Sénégas & Avery, 1998*), and microfauna from the deposits has been used to argue a terminal Miocene or earlier Pliocene age for the deposits (5–4 Ma, *Sénégas & Avery, 1998*; 5.4–05 Ma, *Gommery et al., 2008a*).

The HOPE (Human Origins and Past Environments) project, a collaboration of French and South African researchers based out of the Ditsong National Museum of Natural History, worked at the site from 2001. They attempted to align the UCAE 'loci' on *Cooke*'s (*1991*) map with those observed in the field (*Sénégas et al., 2002*; *Thackeray et al., 2008*). From 2006 HOPE transformed into the HRU (HOPE Research Unit), conducting regular survey and excavations at Bolt's Farm. As a result, several previously undiscovered sites were described (*Gommery et al., 2012*). In order to expose the bone rich *in situ* breccias, detailed excavation of several unstudied deposits (Pit 14, Brad Pit A & B, Milo A & B) were undertaken. An updated map was presented in *Thackeray et al. (2008)*, which included the re-identified deposits from *Sénégas et al. (2002)* and used names rather than the original UCAE Pit numbers: Pit 7 renamed Bridge Cave, Pit 11 renamed X Cave, Pit 14 (incorrectly listed as Pit 15) is renamed Aves Cave and Pit 3 renamed Cobra Cave. Locations for other UCAE Pits, such as Pit 2 (renamed H Cave), Pit 1, Pit 8 (named Rodent Cave) are also suggested. *Thackeray et al. (2008)* also map a number of 'new' sites in addition to Waypoint 160 and Alcelaphine Cave, including Dom's Site, Machine Cave, X Cave and Y Cave.

*Gommery et al. (2012)* built on this research when describing another series of 'new' sites, including a sequence north of Pit 23 called Brad Pit A-C, a series west of Pit 6 called

Milo's Pit A and B, Brigitte Bones A and B, and Carnivore Pit. Further to the northwest another new locality is designated Franky's Cave (*Gommery et al., 2012*). *Gommery et al. (2014)* present a simplified map of the Klinkerts property pits (excluding new localities Brigitte Bones, Dom's and Brad Pit C).

*Monson, Brasil & Hlusko (2015)* attempted to clarify issues around the naming of pits through a historical summary, along with the accession of taxa from the previously unreported New Cave and Jackal Cave. While the authors included a summary table with alternative names for the original pits recorded in 1947, sites since discovered or with material not accessioned at UCMP (e.g., Waypoint 160) were not included.

The history of staggered research at Bolt's Farm spanning eight decades has created a number of issues regarding the consistency of naming practices across the site, with some pits acquiring two names, or being 'double discovered'. This paper aims to provide clarity and rectify these issues of misidentification. Our intent is to create a transparent scheme, advocating for a return to the original naming practices of the site initiated by Camp and Peabody, while also producing a new georectified map to assist in ongoing research at the site (Fig. 2).

## METHODS

Field work was undertaken as part of South African Heritage Resource Agency Permit ID 866, Ref No. 9/2/233/0032.

### Aerial imagery, site survey and GIS

High-resolution aerial imagery was obtained using an eBee senseFly drone. Imagery was processed using Agisoft PhotoScan Pro 1.16 and Georectified on to the South African Coordinate System (Hartebeesthoek 94/ Lo27, EPSG:2052, SA 2010 GEOID), and later converted to World Geodetic System (WGS) 84 Universal Transverse Mercator (UTM) Zone 35S for convenience. Survey control points were established at twelve locations across the site (SOM F3). These were then exploited for a feature based foot survey of the landscape using a Leica GPS1200+ Differential Global Positioning System (DGPS), which enabled sub-centimetre accuracy of surveying positions. This recorded the location of all pits, caves, trenches, historical structures and geological outcrops. DGPS survey was processed with Leica Geo Office and exported to ascii format. Both the Aerial imagery and survey data were imported into ESRI software, ArcMap and ArcScene 10.4. Historical imagery (Peabody's map and the later maps of (*Cooke, 1991*; *Sénégas et al., 2002*; *Thackeray et al., 2008*; *Gommery et al., 2012*) were georectified on to the aerial imagery, allowing for a direct comparison between our new data and the previous maps (Fig. 3). The raw DGPS data (converted to UTM 35s) has been provided, in addition to drone aerial imagery, and our new georectified site map, made available via figshare.

### Faunal analysis

The Bolt's Farm faunas are curated across three international institutions. The University of California Expedition sample is now curated at the University of California Museum of Paleontology (UCMP) at the University of California, Berkeley (*Cooke, 1991*; *Cooke,*

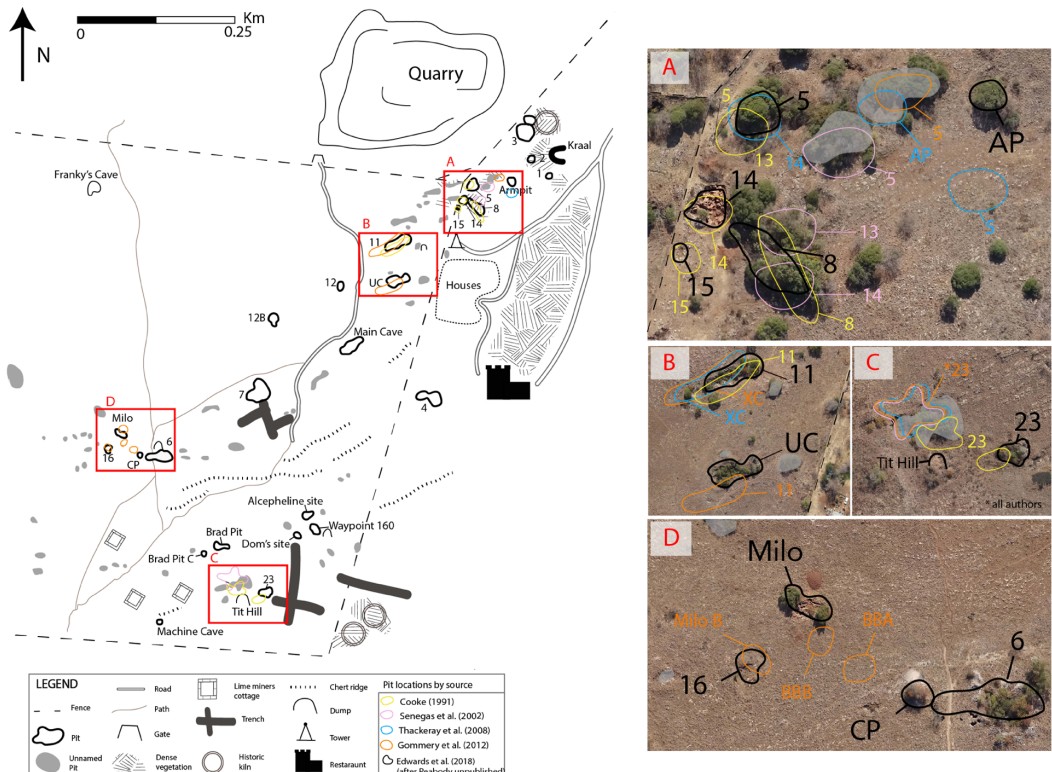

**Figure 3  New map of Bolt's Farm with areas of pit location error highlighted A–D.** Colours represent errors by source. (A) Errors pit locations in the 'Aves Cave Complex' including Pit5, 8, 13, 14, 15, Arm Pit. (B) Errors in location of Pit 11 and U Cave. (C) Misidentification of Pit 23 (D) Misidentification of Pit 16 as new site Milo B and errors in the location of BBA and BBB

*1993*; *Monson, Brasil & Hlusko, 2015*). Decades of intermittent processing and cataloguing has produced a substantial sample across most of the pits across the Bolt's Farm complex. Direct evaluation of specimens to establish primary identification were made in reference to the extensive body of published descriptions of the UCMP and larger South African record, an extensive database of measurements, photographs, and notes on South African fossils and an unpublished summative manuscript on the UCMP collections provided by HBS Cooke (HBS Cooke, pers. comm., 2008). These collections were studied directly by one of us (JWA) during two data collection periods in 2007 and 2012 in collaboration with Dr. Alan Shabel (Department of Integrative Biology, UC Berkeley).

Two South African institutions (Evolutionary Studies Institute, University of the Witwatersrand, Johannesburg; Ditsong National Museum of Natural History, Pretoria) are repositories for some Bolt's Farm specimens and have been regularly studied by JWA over the course of the last 15 years, and were evaluated specifically for this study during field seasons in 2015–2017. Fossils described from recent excavations at Bolt's Farm (e.g., those conducted since the UCAE) were not available for direct study, and any reference to these fossils in our review of the biochronologically relevant taxa comes from published

**Table 2** **Summary of localities discovered subsequent to UCAE mapping.** GPS coordinates as first published and where possible, new accurate DGPS data.

| New Locality 1996–2016 | WGS 84 position and reference | SA Hartebeesthoek 94/ Lo27 (This publication) | UTM -35 Location (This publication) |
|---|---|---|---|
| Waypoint 160 | S26°02′02.0″E27°42′50.0″(*Sénégas et al., 2002*) | −71441.694Y 2880778.398X | 7120373.913N 571413.117E |
| Brad Pit A and B | S26°02′02.8″E27°42′44.2″and S26°02′02.6″E27°42′43.8″(*Gommery et al., 2012*) | −71285.624Y 2880805.139X | 7120347.183N 571257.110E |
| U Cave | S26°1′49.20″E27°42′54.25″(*Thackeray et al., 2008*) | −71570.450Y 2880386.746X | 7120765.408N 571541.822E |
| Brigitte Bones A | S26°01′57.4″E27°42′38.6″(*Gommery et al., 2012*) | Not located from provided coordinates | Not located from provided coordinates |
| Brigitte Bones B | S26°01′57.6″E27°42′38.2″(*Gommery et al., 2012*) | Not located from provided coordinates | Not located from provided coordinates |
| Alcephaline Site | S26°02′00.8″E27°42′49.0″(*Sénégas et al., 2002*) | −71428.251 2880756.014 | 7120393.54N 571398.117E |
| Franky's Cave | S26°01′44.6″E27°42′36.6″(*Gommery et al., 2012*) | −71087.901 2880229.732 | 7120922.934N 571057.728E |
| Carnivore Pit | S26°01′57.8″E27°42′39.1″(*Gommery et al., 2012*) | −71167.72 2880654.998 | 7120497.072N 571140.817E |
| Dom's Site | S26°02′02.0 E27°42′48.8″(*Thackeray et al., 2008*) | −71413.037 2880786.441 | 7120366.659N 571385.409E |
| Machine Cave | S26°02′06.6″E27°42′40.4″(*Thackeray et al., 2008*) | −71191.354 2880923.537 | 7120228.221N 571161.893E |

literature—with the exception of the Milo's A suids which were examined earlier (*Gommery et al., 2012*).

# RESULTS

## Combining legacy maps and accurate spatial data

Table 1 shows the Peabody map localities and associated modern pit names and new DGPS coordinates for known locations. Note that some pits from 1947 have now been re-identified but were listed as 'new discoveries' by subsequent publications (*Sénégas et al., 2002*; *Thackeray et al., 2008*; *Gommery et al., 2012*). Table 2 presents a list of new locales, from work conducted between 1996–2016 which have published fauna associated with the deposits (*Sénégas & Avery, 1998*; *Sénégas, 2000*; *Sénégas et al., 2002*; *Thackeray et al., 2008*; *Gommery et al., 2012*; *Gommery et al., 2014*; *Gommery et al., 2016*; *Pickford & Gommery, 2016*).

Accurate locations of all pits across the Klinkerts and Greensleeves properties are presented in Fig. 2. These data have been overlain with a georectified version of Peabody's original map, *Cooke (1991)*'s interpretation of this map, and subsequent publications which relocated pits and announced new localities; *Sénégas et al. (2002)*, *Thackeray et al. (2008)* and *Gommery et al. (2012)* with discrepancies and clarification of complicated areas shown in Fig. 3.

Importing and georectifying Peabody's original map with our DGPS data and published maps from 1991–2012 identifies discrepancies in four areas (Table 1; Figs. 3A–3D). Three

of these relate to ambiguity in the first published map (*Cooke, 1991*), from which all subsequent maps until now were produced. Firstly, the precise locations of Pits 5 and 13–15 are not easily discernible (Fig. 3A). The location of Pit 11 is correctly identified by *Cooke (1991)* (Fig. 3B). The designation of Pit 23 is placed between two localities whereas Peabody labels Pit 23 as the more easterly of the two pits (Fig. 3C). Through georectification of the original map and archival research (SOM SF2, SF3) we have determined Pit 23 to be the more easterly of the two pits, however it has been continually misidentified in the literature. The location of Pit 16 is cut off the map, allowing for this to be re-discovered as a new site more than twenty years later (Fig. 3D). Without direct comparison with the original Peabody map it is impossible to interpret these complex areas on Cooke's map.

*Sénégas et al. (2002)* published a map following the *Cooke (1991)* version along with GPS coordinates for Pits 3–7, 9, 11–15 and 23 (Table 1). This new map features 'Breccia outcrop' from *Cooke*'s (*1991*) map and 'new' locations Waypoint 160, Alcelaphine Site and the Femur Dump (*Sénégas et al., 2002*; *Gommery, Sénégas & Thackeray, 2008b*). While the latter is present as 'Tit Hill' on Peabody's map, it was not copied over by *Cooke (1991)* and ambiguity in this region led to misidentification of Pit 23 (Fig. 3B). Most of the locations reported in *Sénégas et al. (2002)* plot close to identifiable pits on new aerial imagery, with a few exceptions. Firstly, 'Breccia outcrop' plots directly adjacent to Pit 6, making it possible that a breccia dump was mistakenly logged as an outcrop. Digital comparison of both maps (*Cooke, 1991*; *Sénégas et al., 2002*) show that the 'Breccia outcrop' locations do not correlate spatially. There was uncertainty regarding which deposit represented Pit 12, resulting in the creation of Pit 12A and 12B. Moreover, the location for Pits 5 and 13, while being associated with a pit on aerial imagery is not where the original Pits 5, 13 and 14 are located (Fig. 3A). Archival research of original field notebooks at the UCMP showed Pit 13 to be a dump associated with Pits 5 and 14 (SOM SF5), which is not clear from looking at either the Peabody or *Cooke (1991)* map.

*Thackeray et al. (2008)* present an overview of research at Bolt's Farm and include an updated map with several new localities along with GPS coordinates. Plotting these coordinates on georectified aerial image shows several inconsistencies with the original mapped pits (Fig. 3). While Pit 14 was correctly identified as Benchmark Pit, coordinates given match those at Pit 8 (Fig. 3A). Pit 5 was placed more than 20m away from the original mapped pit. They map in a pit which is identified as Pit 13 and given the name Arm Pit; however, as stated above, archival research reveals Pit 13 was a dump. Ultimately, Arm Pit does correspond to a real world location and moving forward should continue with this name without the designation of Pit 13 (Fig. 3A). GPS coordinates show that Pit 11 is incorrectly identified as a new site, X Cave while U Cave located to the south is labelled Pit 11 (Fig. 3B). Following *Cooke (1991)*'s map and *Sénégas et al. (2002)* Pit 23 is incorrectly identified (Fig. 3C).

*Gommery et al. (2012)* present nine newly discovered localities with GPS coordinates. While many of the discoveries are legitimate with coordinates that plot close to identifiable pits (Brad Pit A–C, Alcelaphine Site, Dom's Cave) others are misidentifications of old sites or there are issues with the coordinates. Several misidentifications continue through the literature including Pit 11, Pit 23, Pit 14 and Pit 5 (Fig. 3). The new sites Milo A and Milo

**Table 3** List of Pits with maximum and minimum depositional ages as indicated by biochronologically informative species.

| Pit Number | Max Age | Min Age |
| --- | --- | --- |
| Pit 1 | <2.33 Ma | 0.78 |
| Pit 2 | NA | NA |
| Pit 3 | <2.33 Ma/1.89 Ma | 0.78 |
| Pit 4 | <2.33 Ma | NA |
| Pit 5 | <2.33 Ma | NA |
| Pit 6 | <2.33 Ma | 0.78 |
| Pit 7 | 4.4 Ma | 2.5 Ma (2.0 Ma) |
| Pit 8 | NA | NA |
| Pit 10 | <3.7 Ma | NA |
| Milo A | 3.03–2.58 | >1.95 |
| Pit 11 | <2 Ma | NA |
| Pit 14 | 3.03–2.58 | >1.95 |
| Pit 15 | NA | NA |
| Pit 16 | <2.33 Ma | 0.99 Ma |
| Pit 23 | 3.03–2.58 | >1.95 |
| Pit 25 | <2.33 Ma | 0.78 |
| Jackal Cave | NA | NA |
| New Cave | <2.33 Ma | NA |
| Waypoint 160 | <5.0 | NA |
| Brad Pit | N/A | N/A |

B correspond to localities mapped by the UCAE in 1947: 'Bushman outcrop' and Pit 16 respectively (Fig. 3D). Using both supplied coordinates and overlaying our georectified map, we were unable to align Brigitte Bones A or B with any identifiable pits (Fig. 3D).

Some of the issues raised here were addressed by *Pickford & Gommery (2016)* who used, but did not publish in full, Peabody's original map. Access to this allowed them to identify and correct many errors made especially in the area they have called the 'Aves Cave Complex'. However, while Pits 8, 14 and 15 are correctly identified Pit 5 is incorrectly labelled Pit 13. Direct comparison with the map published in *Pickford & Gommery (2016)* was not possible due to small size of their map, which limited accurate georectification.

## Biochonologically Significant Bolt's Farm Fauna

A full description of the biochronologically-informative faunas from the Bolt's Farm localities described to date is provided in full in our SOM (Text S1) and the summed results of our evaluation are presented in Table 3. We wish to emphasise that the faunal data and descriptions provided here and within supplementary online material, while reflecting a substantial advance over prior taxon-focused or summative publications on the Bolt's Farm fossil faunas, is only inclusive of specimens broadly relevant for establishing biochronological interpretations of the pit deposits. The descriptions and discussion should not be taken as a comprehensive description or listing of taxa from these deposits across these institutions.

There is insufficient faunal data from Pits 2, 8, 15, 17, Jackal Cave and Brad Pit A and B to establish a biochronological age bracket for these deposits. The majority of the described Bolt's Farm localities were deposited after 2.33 Ma given the regular recovery of *Equus* specimens that must postdate the entry of the genus into Africa (Table 3; *Geraads, Raynal & Eisenmann, 2004*). A probable minimum depositional age boundary of 0.78 Ma can be established for Pits 1, 3, and 25 by the occurrence of the extinct bovid *Antidorcas recki*, which disappears from South African deposits after the formation of Elandsfontein (*Klein et al., 2007*; *Braun et al., 2013*). Pit 16 contains extinct three-toed horse (*Eurygnathohippus*) and was likely deposited prior to 0.99 Ma (SOM Text S1). Pits 4, 5, and New Cave lack fauna that can restrict the minimum depositional age.

Only the Pits 7, 10, 14, 23, Waypoint 160 and Milo's A deposits contain fauna that may have been deposited prior to 2.33 Ma. The recovery of an extinct elephant (*Elephas*) from Pit 7 suggests a maximal depositional age of 4.4–2.5 Ma (potentially extending to 2.0 Ma; SOM Text S1); however, as noted above the provenance of the specimen within the deposits is unknown and a recent U-Pb age indicates some flowstones in the cave formed <∼1.8 Ma (*Pickering et al., 2018*) As such, an in depth study of the Pit 7 stratigraphy and potential associations of the specimen will be necessary to to establish a robust chronology for this location. The Pit 10 deposits contain the type specimen of the herpestid *Ictonyx bolti* (subsequently subsumed into *Prepoecilogale bolti Cooke, 1985*) known only to occur in the late Pliocene (∼3.7–2.5 Ma) from northern and eastern African deposits (SOM Text S1). The Pit 14, 23 and Milo's A deposits all contain Stage I *Metridiochoerus andrewsi* craniodental remains that are morphologically analogous to those recovered from the Makapansgat Member 3 deposits (3.03–2.58 Ma) (*Partridge, 1973*; *Herries, Curnoe & Adams, 2009*; *Herries et al., 2013*). This may reflect a similar maximal depositional age; however, the limits of the South African suid record mean that at present we can only infer deposition of these specimens prior to 1.95 Ma (SOM Text S1). Finally, although Waypoint 160 has been previously suggested to date to after the Langebaanweg E Quarry deposits (∼5.2 Ma; *Roberts et al., 2011*) and prior to the Makapansgat Member 3 deposits (3.03–2.58 Ma), as noted above and in SOM Text S1, without an established FAD or LAD for *Euryotomys bolti* and the recent identification of *Panthera* cf. *leo*, such a Pliocene age is not clearly supported by the fauna. Equally, recent U-Pb ages suggest flowstones in the cave formed at <∼2.3 Ma, supporting the notion that at least some of this deposit is Early Pleistocene (*Pickering et al., 2018*).

## DISCUSSION

The extensive history of research at Bolt's Farm has yielded a substantial and diverse faunal sample from the known localities. The palaeontological significance of Bolt's Farm has lagged behind that of other South African deposits due to the divided curation of materials from across the deposits, the sporadic history of excavation, and confusion over location and nomenclature of specific pits.

The combination of several different teams working at Bolt's Farm through the decades, often with significant time between excavations and collections, and the disturbance of

many of the deposits by lime mining has cumulatively lead to the present situation of multiple names for individual deposits and ambiguity as to the exact location of a number of the pits. While attempts have been made to reconcile disparity between the naming of deposits and faunal assemblages (*Monson, Brasil & Hlusko, 2015*) and to build new naming strategies for the pits (*Pickford & Gommery, 2016*), the lack of an overarching approach focused on the accurate spatial identification of original and recently discovered pits has only added to the confusion.

By digitally overlaying Peabody's original map (*Monson, Brasil & Hlusko, 2015*) and subsequently published maps (*Cooke, 1991*; *Sénégas et al., 2002*; *Gommery et al., 2012*) with new aerial imagery and survey data, we are able to recognise pit misidentifications and errors with naming (Fig. 3). Spatially accurate mapping of palaeontological sites is crucial for ongoing work, especially palaeomagnetic and Uranium-Lead (U-Pb) dating, which both require secure stratigraphic contexts. In addition, the provision of 3D surveying benchmarks across the site means that all future fossil and geological samples can be recorded *in situ* and to a high degree of spatial accuracy, thereby resolving the issue of contextual and provenance problems. The work presented here is the first of its kind conducted on the site since 1947–1948, reinforcing the need for these types of surveys to be conducted, both in the context of ongoing excavation and with the analysis of historical collections.

Given our comparison of Peabody's original map with published material and the errors in naming identified (Fig. 3), we strongly recommend that all pits be referenced by their number or original title where possible (Table 1; Fig. 2). For the majority of pits across the site this is the numerical designator assigned during the UCAE (e.g., Pits 1–23). However, for all truly new sites subsequently discovered (e.g., Waypoint 160), their first published name should be used to prevent any further confusion. Since no material was recovered from "Bushman outcrop" it should henceforth be known by the first name associated with published faunal material "Milo". Additionally, due to the questionable name attributed Pit 3 by the UCAE, the numerical designator (3) or new HRU name (Cobra Cave) is favoured (Table 1).

Biochronological assessment of the faunal specimens from the Pits suggests that parts of the Bolt's Farm complex may be the oldest in the Blaubank Stream Valley, possibly forming as early as the mid- (e.g., Pits 7 and 10) or late (e.g., Pits 14, 23 and Milo's A) Pliocene, and therefore prior or contemporaneous with the formation of the Makapansgat Member 3 deposits (3.03–2.58 Ma; *Herries et al., 2013*). Recent U-Pb ages for flowstones at some of these deposits (Pit 7, Pit 14, Waypoint 160; *Pickering et al., 2018*) may help to further refine or constrain these ages when combined with in depth stratigraphic interpretation and other chronological methods. These ages appear to suggest that deposits within the Cradle are all younger than ~3.2 Ma. With a combined record that may span over 2 Ma of deposition, Bolt's Farm represents—alongside Sterkfontein—one of few site complexes to cover such a long span of time in the Cradle region, providing a rare opportunity for more detailed comparisons of the fauna from these different localities through time (*Pickering et al., 2018*; *Herries et al., 2018*).

Additionally, within the Cradle it is unusual to have an extensive site complex like Bolt's Farm that is devoid of hominin specimens, and a small non-hominin primate sample, in such close proximity to well-known hominin- and primate-bearing sites (e.g., Sterkfontein, Swartkrans, Rising Star). There are many potential reasons why hominins or primates may not occur within the Bolt's Farm deposits which warrant mention. There are numerous references within the original field notes of Camp to australopithecine and "ape man" remains from Pit 3 (SOM SF6. SF7. SF8); however, these specimens are not known to have been subsequently catalogued within any current collections. It is possible that these specimens were incorrectly identified in the field (e.g., reclassified as non-hominin primate or other mammal remains), or that they were accidentally integrated into other fossil samples during the removal of Bolt's Farm materials which saw them organised and packed at the Ditsong National Museum of Natural History prior to export. We can establish that some specimens were simply never accessioned. For example, while Pit 3 is the only location from which a single stone tool is known to have been recovered; however, Camp's notes provide insight citing that he "scraped out 10–15 blades and gave them to the (Bolt) sisters" (SOM SF9). He goes on to list artefacts "thin blades, quartz chips. One core of chert and some slate artefacts"; none of these artefacts are known today. Equally, variable taphonomic processes exert a strong mediating role in faunal assemblage composition (*Brain, 1981*; *Pickering, 1999*; *Adams, 2006*; *Pickering et al., 2004*; *Val & Stratford, 2015*) and the taphonomic histories of these Pits have not yet been addressed (excepting Pit 23; see *Brain, 1981*). Ultimately, it is important to highlight that a bias towards excavating and analysing the well-known hominin fossil sites located nearby may be distorting our perception of how regularly hominins, primates and archaeological materials were integrated into the Cradle localities. In this respect, the Bolt's Farm Pits may be typical of penecontemporaneous deposition across the region in representation of fauna.

## CONCLUSIONS AND FUTURE RESEARCH

In the more than 80 years since Broom first prospected at Bolt's Farm, continued research has proven the value of the site to yield important palaeontological remains, the summed sample of which indicates an extensive depositional history that has been suggested to date back into the Pliocene.

Bolt's Farm differs significantly from other sites in the Cradle in two ways. Firstly, while palaeokarst features are commonplace throughout the Cradle, most fossil bearing sites are either caves (e.g., Sterkfontein) or single palaeokarst deposits (e.g., Malapa). It is unprecedented to have such a high density of fossil bearing palaeokarst deposits and active caves in a small area, as is the case at Bolt's Farm. Additionally, biochronology suggests there is significant temporal variation within, and between, the more than twenty known localities across the site. The unique conditions which have led to the preservation of so many palaeokarst remnants and caves is inherently linked to the geology observed at the site, requiring further research to fully disentangle.

It is critical to the next stage of research at Bolt's Farm that all areas be accurately mapped and a uniform naming scheme be settled on. As a result, the detailed survey

provided here seeks to clarify the naming issues and we present the first new map of the site in more than 70 years. Our study highlights the importance of field survey paired with high-resolution spatial mapping and drone survey, as our new map and site surveying control points allow the historical fossil collection to be accurately placed within its original context. The continued use of 3D data collection methodologies at the site will rectify some of the problems researchers have encountered. Although the site has been disturbed by mining activities and some contexts destroyed, the importance of this information is only being realised as new methods enable these distinct areas to be dated. While additional biochronological dating (after full description of more recently excavated faunas) and absolute dating methods will provide clarification of the age of deposits, spatial aids provided here should be adopted by researchers continuing to excavate at Bolt's Farm, to ensure an accurate spatial and contextual record of all finds from this key palaeontological site in the Cradle.

## ACKNOWLEDGEMENTS

The authors wish to thank Dr. Robert Anemone and one anonymous reviewer for helpful suggestions which improved this manuscript. The authors wish to thank Stepheny Potze and Lazarus Kgasi for access to the site and assistance with fieldwork, Stephany Potze was the SAHRA permit holder during initial stages of field work Dominique Gommery and Lazarus Kgasi SAHRA permit holders during surface survey; Ditsong National Museum of Natural History; Pat Holroyd for access to original notes and collections of UCAE, UCMP ; Tara R. Edwards thanks Tesla Monson for assistance, support and helpful discussion, Justin W. Adams thanks Alan Shabel for assistance, support and expertise at UCMP; J Gaylord, landowner of Greensleeves. We thank Norbert Plate of iQlaser (http://www.iqlaser.co.za) for his time, equipment and provision of aerial imagery. Elevation data available from USGS.

### Funding

Funding received from Australia Research Council (grant FT120100399 to Andy I.R. Herries and DP170100056 to Andy I.R. Herries and J.W. Adams), the La Trobe University Humanities and Social Science Internal Research Grant Scheme (#2015-1-HDR-1 to Brian J. Armstrong and #2017-1-HDR-0009 to Tara R. Edwards) and National Research Foundation African Origins Platform (grant AOP150924142990 to Robyn Pickering). The funders had no role in study design, data collection and analysis, decision to publish, or preparation of the manuscript.

### Grant Disclosures

The following grant information was disclosed by the authors:
Australia Research Council: FT120100399, DP170100056.
The La Trobe University Humanities and Social Science Internal Research Grant Scheme: #2015-1-HDR-1 to BJA, #2017-1-HDR-0009.
National Research Foundation African Origins Platform: AOP150924142990.

## Competing Interests

The authors declare we have no competing interests.

## Author Contributions

- Tara R. Edwards conceived and designed the experiments, performed the experiments, analyzed the data, contributed reagents/materials/analysis tools, prepared figures and/or tables, authored or reviewed drafts of the paper, approved the final draft.
- Brian J. Armstrong conceived and designed the experiments, performed the experiments, analyzed the data, contributed reagents/materials/analysis tools, prepared figures and/or tables, approved the final draft.
- Jessie Birkett-Rees, Alexander F. Blackwood performed the experiments, analyzed the data, contributed reagents/materials/analysis tools, prepared figures and/or tables, approved the final draft.
- Andy I.R. Herries conceived and designed the experiments, authored or reviewed drafts of the paper, approved the final draft.
- Paul Penzo-Kajewski performed the experiments, analyzed the data, contributed reagents/materials/analysis tools, approved the final draft.
- Robyn Pickering authored or reviewed drafts of the paper, approved the final draft.
- Justin W. Adams conceived and designed the experiments, performed the experiments, authored or reviewed drafts of the paper, approved the final draft.

## Field Study Permissions

The following information was supplied relating to field study approvals (i.e., approving body and any reference numbers):

South African Heritage Resource Agency approved the study (Permit ID 866, Ref No. 9/2/233/0032).

## Data Availability

Edwards, Tara (2018): Edwardsetal_BFSOM. figshare. Fileset. https://doi.org/10.26181/5bce7a0611c4b.

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
