# Peer review of "Combining legacy data with new drone and DGPS mapping to identify the provenance of Plio-Pleistocene fossils from Bolt’s Farm, Cradle of Humankind (South Africa)"

_PeerJ, doi:10.7717/peerj.6202_

## Round 0.1 · original submission · Minor Revisions

I have received 2 very positive reviews of your paper on Bolt's Farm, both of which see the need for only minor revisions. Both offer helpful annotations that I urge you to read and incorporate. Reviewer 1 (Dr. Anemone) requests additional citations to contextualize your work and to give credit to others using similar methods in paleoanthropology and vertebrate paleontology. I support this recommendation. I anticipate that no further review will be necessary when we receive your revised manuscript.

·

Basic reporting

This article follows a recent trend in which paleoanthropologists and vertebrate paleontologists are increasingly using sophisticated tools, methods, and datasets from the Geographic Information Sciences (GISciences) in their explorations for and analyses of fossil-bearing deposits around the world. It would be appropriate for the authors to include citations to some of this recent work in paleoanthropology and vertebrate paleontology (published in journals like JHE, JVP, AJPA, Evol Anthro and others), and to thereby place their work within this historical context and evolving tradition within our disciplines. (Archaeologists have been using sophisticated geospatial tools for much longer than paleoanthropologists). The writing is clear and unambiguous, although many small suggestions and edits have been included on the uploaded pdf, and these will improve the tone, clarity, and the sense of the manuscript. The Figures and Tables are relevant and high quality and appropriately labeled for the most part. Figure 1 depicts what looks like a DEM of South Africa, yet the color ramp is not defined. Figure 1 also includes a topographic map yet doesn’t label the contour interval. Please include the map projection or UTM zone for Figure 2 below the scale bar.

Experimental design

The design of the research is clear, its aims are well within the scope of PeerJ, and its methods are rigorous, utilizing state of the art geospatial techniques to clarify long-standing confusion as to the location of fossil sites at Bolt’s Farm. The Methods are described in sufficient detail and the Supplementary Data are relevant and provide added value to this work (esp. the detailed faunal lists and notes concerning biochronology).

Validity of the findings

The findings are valid and help to clarify long-standing confusion as to the identity and location of fossil-0bearing sites at Bolt’s Farm, resulting from a long and episodic history of work there by different teams separated by long temporal gaps. The authors have done the field a service by restudying the collections from Bolt’s Farm at different institutions on different continents. The biochronological conclusions are necessarily limited in their scope and finality, but this is mostly a result of limitations in the data themselves rather than a problem with the current research design or execution.

Reviewer 2 ·

Basic reporting

Some minor edits are suggested on the pdf itself in terms of grammar and wording. It is a well written paper with relevant information and high quality figures that will certainly provide guidance for future scientists that are interested in field work at this site or the fauna that has been recovered from there.

Experimental design

All of the previous maps of the site, as well as any potential sources of information (like field journals) were accessed for this study, giving the sense that this truly is an exhaustive description of the site. Methods for how various maps were combined with current aerial photography were clear.

Validity of the findings

The recommendation of using the original names of the pits when possible is useful and as a great example of how research at several Cradle sites could move forward with more information that is better translated across research groups. Errors in naming are likely seen at many sites that have had different research projects over the years, and this paper serves as a good example of potential “best practices” in moving forward with rectifying some of the errors. It’s possible other researchers who have worked at the site or will in the future may not agree with the recommendation, but at minimum this paper illustrates which pits are named or renamed by various projects. The data used for the biochronology estimates seem robust and will be interesting in light of new absolute dates coming out of the Cradle.

Additional comments

This paper, “Combining legacy data with new drone and DGPS mapping to identify the provenance of Plio-Pleistocene fossils from Bolt's Farm, Cradle of Humankind (South Africa),” revises the map and names of Bolt’s Farm assemblages, as well as provides maximum and minimum dates via biochronology for those assemblages when possible. The background provided about the excavation history of Bolt’s Farm is valuable, and the tables of site names and related information through time is an extremely important contribution to the field.

Though it is mentioned in the paper that this study did not have a goal of comprehensive faunal lists for each site within Bolt’s Farm, such a contribution would certainly be valuable as a follow-up in the future.

Annotated reviews are not available for download in order to protect the identity of reviewers who chose to remain anonymous.

---

## Round 0.2 · accepted · Accept

Thank you for your attention to reviewer suggestions, the minor "fixes" and quick turn-around.

#